# Robot-Assisted Radiofrequency Ablation Combined with Thermodynamic Simulation for Epilepsy Reoperations

**DOI:** 10.3390/jcm11164804

**Published:** 2022-08-17

**Authors:** Yu-Chi Wang, Mei-Yun Cheng, Po-Cheng Hung, Cheng-Yen Kuo, Hsiang-Yao Hsieh, Kuang-Lin Lin, Po-Hsun Tu, Chieh-Tsai Wu, Peng-Wei Hsu, Kuo-Chen Wei, Chi-Cheng Chuang

**Affiliations:** 1Department of Neurosurgery, Chang Gung Memorial Hospital at Linkou, Taoyuan 333, Taiwan; 2School of Medicine, Chang Gung University, Taoyuan 333, Taiwan; 3Department of Neurology, Change Gung Memorial Hospital at Linkou, Taoyuan 333, Taiwan; 4Division of Pediatric Neurology, Chang Gung Children’s Hospital, Taoyuan 333, Taiwan; 5Department of Pediatrics, Chang Gung Children’s Hospital, Taoyuan 333, Taiwan; 6Department of Neurosurgery, New Taipei Municipal Tu Cheng Hospital, Chang Gung Medical Foundation, Taipei 236, Taiwan

**Keywords:** epilepsy surgery, radiofrequency ablation, robot-assisted, computer simulation

## Abstract

Repeat craniotomies to treat recurrent seizures may be difficult, and minimally invasive radiofrequency ablation is an alternative therapy. On the basis of this procedure, we aimed to develop a more reliable methodology which is helpful for institutions where real-time image monitoring or electrophysiologic guidance during ablation are not available. We used simulation combined with a robot-assisted radiofrequency ablation (S-RARFA) protocol to plan and execute brain epileptic tissue lesioning. Trajectories of electrodes were planned on the robot system, and time-dependent thermodynamics was simulated with radiofrequency parameters. Thermal gradient and margin were displayed on a computer to calculate ablation volume with a mathematic equation. Actual volume was measured on images after the ablation. This small series included one pediatric and two adult patients. The remnant hippocampus, corpus callosum, and irritative zone around arteriovenous malformation nidus were all treated with S-RARFA. The mean error percentage of the volume ablated between preoperative simulation and postoperative measurement was 2.4 ± 0.7%. No complications or newly developed neurologic deficits presented postoperatively, and the patients had little postoperative pain and short hospital stays. In this pilot study, we preliminarily verified the feasibility and safety of this novel protocol. As an alternative to traditional surgeries or real-time monitoring, S-RARFA served as successful seizure reoperation with high accuracy, minimal collateral damage, and good seizure control.

## 1. Introduction

Epilepsy is considered to be a disease of the brain which is defined by any of the following conditions: (1) at least two unprovoked (or reflex) seizures occurring > 24 h apart; (2) one unprovoked (or reflex) seizure and a probability of further seizures similar to the general recurrence risk (at least 60%) after two unprovoked seizures, occurring over the next 10 years; (3) diagnosis of an epilepsy syndrome. Anti-epileptic drugs (AEDs) are the first line treatment for controlling neuronal kindling in the brain. However, around one third of patients have drug-refractory epilepsy, which is defined as the failure of adequate trials of at least two antiepileptic drugs that are appropriately chosen, used, and tolerated [1,2].

Surgery has been shown to be an effective treatment option for patients with drug-refractory epilepsy. Various seizure surgeries such as anterior temporal lobectomy (ATL) and selective amygdalohippocampectomy (SAH) were used to treat focal seizure arising from the temporal lobe [3]. The main idea of these surgical interventions is that selective destruction of epileptogenic foci or critical nodes in epileptogenic networks may result in seizure control. However, generalized seizures with multifoci are rarely controlled with focal surgical lesioning. Among them, Lennox–Gastaut syndrome (LGS), which is a clinical diagnosis characterized by multiple seizure types, cognitive impairment, and a complicated electroencephalogram (EEG) pattern, remains a surgical treatment challenge [4]. Corpus callosotomy (CC) has been traditionally employed as a palliative treatment option for LGS patients with drop attacks [5]. Postoperative freedom from seizures or a worthwhile reduction for months to years have been reported [3]. However, when the seizures recur, repeated craniotomies potentially increase the risk of comorbidities due to adhesion and manipulation [6].

Several minimally invasive procedures have been developed to treat seizures with the least collateral damage [7]. Epilepsy surgeries in combination with stereotactic methods such as robotic stereotactic assistance (ROSA, Zimmer Biomet, Stamford, CT, USA) are used to treat different types of epilepsy with high accuracy. Stereotactic electroencephalography (SEEG)-guided radiofrequency thermocoagulation (RFTC) has been used for patients whose epileptic foci are not eligible for resection [8,9]. However, the only real-time imaging system to monitor ablation is laser interstitial thermal therapy (LiTT) [10,11], which is not available in East Asia.

To allow safe coagulations in rich vascular environments intracranially, the electrode first needs to reach the targeted anatomical structure with a good geometric accuracy in order to be appropriate for RFTC [12]. Second, the maximal lesion volume may be achieved by using parameters adjusted according to the impedance for each coagulation site. However, parameters of radiofrequency current were adapted based on surgeons’ experience to optimize the size of the lesion, whereas the lesion size obtained using standard fixed parameters is usually smaller [8]. Therefore, computational calculation to simulate tissue temperature during ablation with specific parameters might be a solution to make surgeons more understanding of thermodynamic status prior to operation, since a programing platform was shown to be accurate in the demonstration of electromagnetic fields on animal models [13]. In addition, a computational model was used to estimate the temperature profile in the human brain resulting from exposure to various radiofrequency field parameters in a previous study [14].

In order to optimize postoperative seizure control, the accurate site of ablation is the priority. Meanwhile, to minimize collateral damage, a predictable lesion size is the solution. Therefore, we sought to use computer simulation for preoperative parameter setting, synergizing intraoperative robot-assisted radiofrequency ablation, here abbreviated S-RARFA, on deep-seated intracranial structures as seizure control reoperations for patients not eligible for repeated craniotomy.

## 2. Materials and Methods

This study was approved by the Chang Gung Memorial Hospital Institutional Review Board. Patients were recruited who met the inclusion criteria of recurrent drug-resistant epilepsy, defined as refractory to trials of at least two appropriately chosen anticonvulsant medications, and who had seizure surgery before. All patients had a noninvasive epilepsy study of a 3-day scalp EEG to localize the epileptic zone, and a positron emission tomography brain magnetic resonance imaging (PET-MRI) scan for intracranial structure as well as metabolism study. After evaluation and recommendations from a multidisciplinary epilepsy surgery conference that included epileptologists, neuroradiologists, neuropsychologist, and neurosurgeons, all patients consented to undergo S-RARFA to treat their epilepsy.

The epileptic symptoms, categories, and surgical outcomes were assessed with the latest guidance of the International League Against Epilepsy (ILAE) classification [15]. Preoperative and postoperative neuropsychological examinations were performed using the Wechsler Adult Intelligence Scale to evaluate memory function and IQ in adult patients [15,16].

Preoperative thin-slice MRI with and without contrast was performed in various sequences. The attempted ablation site was clearly visualized using T1-weighted MRI. This MRI study was uploaded to the ROSA host, and a plan was created in which the entry and target points were defined, along with a safe pathway that avoided blood vessels seen on contrast imaging. The ablation target, trajectories, and volume were all marked on the system.

The simulation used the methods of coupling electro quasi-static solvers with a thermal simulation in Sim4Life (Zurich MedTech AG, Zurich, Switzerland). A realistic scenario of radio frequency (RF) induced thermal ablation with a unipolar applicator was implemented. This unipolar RF applicator is a thin needle with electrodes at its tip and a grounded plane electrode on the body. The applicator needle was inserted into hippocampus tissue. The following shows the set up in an *EM-LF* simulation and a transient *Thermal* simulation with an electromagnetic (EM) heating source.

For developing a computational anatomic model, iSEG V3.10 (ZMT, Zurich, Switzerland) and Sim4Life V6.2 modeling tools (ZMT, Zurich, Switzerland) were used to create 3D models. A realistic head model of a patient was segmented with white matter, grey matter, and hippocampus by iSEG, and the unipolar electrosurgical unit model was established according to the actual equipment used by Sim4Life modeling tools.

To simulate the Thermal RF treatment to the brain, we used ZMT’s Sim4Life v6.2 Electro Quasi-Static(E-QS) Finite Element Method (FEM) solver. It was used to solve the equation $\nabla \tilde{\epsilon }\nabla \emptyset =0$, (where $\tilde{\epsilon }$ is the complex permittivity, and ∅ is the electric potential) on adaptive structured meshes developed on the model geometries. This solver was chosen as it permits the handling of the low system frequency. The electrical tissue properties (conductivity) for the simulation were set to values at 100 kHz. The normal current was set to 34 V. The heat simulation solver used Penne’s bioheat equation, the most widely used thermal model, which takes into account tissue heat capacity and conductivity [17]. Sensitivity analysis was executed to establish the sensitivity of the temperature distribution within the brain to the thermal properties of the various tissue types. The electric conductivity of grey matter was set to 0.28 S/m, and white matter was set to 0.27 S/m. Sensitivity analysis was performed to establish the sensitivity of the temperature distribution within the brain to the thermal properties of the various tissue types.

Patients were anesthetized in the operative room. Their head was then placed in a Cosman–Roberts–Wells (CRW) frame (Integra, Plainsboro Township, NJ, USA), which was affixed to the operative bed. The ROSA robot was then moved into position, registered with the image using an intraoperative O-arm (Medtronic, Minneapolis, MN, USA) with the fiducial attached on the frame. Using an aligning rod through the robot trajectory guide, a radiofrequency unipolar electrode (Cosman Medical, Burlington, MA, USA), with a 1.6 mm diameter and a 3 mm cautery tip, was passed through the bone. The electrode tip was inserted to the targeted depth first, then moved 3 mm backward axially through the robotic micromove program after each time of ablation (Figure 1A).

Multiple lesions in 2–3 electrode trajectories were made using a radiofrequency lesion generator system (G4, Cosman Medical, Burlington, MA, USA). The lesions were produced around the electrode using a voltage dependent (30–33 V), with 2–6 W power. These parameters were chosen specifically to increase the tissue temperature from 37 °C to 80 °C (±2 °C) in 10–15 s, while the whole thermal ablation persisted for 90 s. This protocol would ideally create a hemisphere at the tip, plus a cylinder with a planned length. The equation of the lesioning volume was therefore defined as:(1)$$V=\frac{1}{2}\pi r^{3}+\pi r^{2}3n$$
where π is 3.14, r is the radius of thermal-ablated tissue, and n is the number of superimposed lesions composing the total axial length (Figure 1A). Actual volumes of ablation, which were defined by a high intensity margin in the postoperative T2-weighted images, were followed by affine registration in the Advanced Normalization Tools software (http://stnava.github.io/ANTs/, accessed on 7 July 2022) [18]. A volumetric comparison between calculated volumes by simulation with volumes measured from actual ablations was performed.

## 3. Results

### 3.1. Ablation Data

Two adults and one child were included in this study. Preoperative computer simulations are demonstrated, among which 90-s ablation was performed. The simulation lesions with 60 °C boundaries were 5.2–5.5 mm in radius, and the radius was substituted into the volume equation. The mean percentage of error between calculated and measured ablation volume was 2.4 ± 0.7% (Table 1).

### 3.2. Case 1. Right Temporal Seizure after ATL

This 28-year-old man with right temporal epilepsy which was refractory to four AEDs underwent right partial ATL sparing the hippocampus at the other institution. The seizures relapsed 5 months later. Three-trajectory RARFA mimicking extent SAH was performed (Appendix A), and he was classified as ILAE class 1a for 2 years after the operation (Figure 1B–D).

### 3.3. Case 2. LGS Treated with Anterior CC

This nine-year old boy had generalized seizure after encephalitis at two years old. LGS was the diagnosis, and he had drop attacks daily, which impaired his quality of life greatly. He underwent incomplete anterior 1/2 CC, and the symptoms relapsed half a year later. Thermal ablation was simulated for a safe margin on the remnant corpus callosum, including the left genu and splenium (Appendix A). Postoperative MRI revealed no injuries on surrounding crucial vessels. Epileptic discharges decreased significantly, and drop attacks remained remitted at 1.5 year postoperative follow-up. He was classified as ILAE class 4, and the number of his AEDs were tapered from 5 to 4 (Figure 2A–C).

### 3.4. Case 3. Intractable Seizure Secondary to Arteriovenous Malformation (AVM,)

This 45-year-old woman had left temporal seizure due to left temporal AVM, which was obliterated significantly after radiosurgery with 3600 cGy. However, her seizure persisted and progressed, even with three AEDs treatment. Her epileptic foci were confirmed ahead and behind of the original AVM through SEEG first, then S-RARFA was performed on two zones. She was classified as ILAE class 1a for 1 year (Figure 3A–C).

### 3.5. Surgical Complication and Outcome

No new neurologic deficits, intracranial hemorrhage, wound infections, or acute postoperative seizures were noted after the procedures in any of the patients. The average length of postoperative hospital stay was 3 days. In two adult patients, a neuropsychologic test demonstrated unchanged memory quotients (MQ) and a small improvement of Full-Scale IQ score (mean change + 4.5 points) half a year after treatment.

## 4. Discussion

Robot assistant systems have increasingly been used in neurosurgery in recent years, and they can facilitate minimally invasive procedures for patients with medically refractory epilepsy. In this study, we demonstrated that, despite lacking real-time monitoring, S-RARFA is a reasonable option through careful preoperative planning and computer simulation. This protocol can provide precise destruction and calculable volumes. From a surgical standpoint, simulation provides visible anatomic references and predicts outcomes. In LiTT, the ablations are monitored in real time using magnetic resonance thermometry. Computer simulation of thermodynamic scenarios can be an alternative method in hospitals where LiTT is unavailable.

The efficacy of RFTC on seizure control was inferior to those of ATL, SAH, or LiTT; the reason for this is the ablation volume and target selection [19]. The superiority of “limited” RFTC on surgical resection reported by Lee et al. is disputed [20,21], and thus we sought to optimize outcomes by planning greater volume ablation on the mesial temporal structure, such as LiTT, to maximize the chance and duration of seizure freedom. The superposition of the trajectories in patient 1 echoed previous studies, showing that ablations should cover more of the amygdala and also include more of the anterior, medial, and inferior temporal lobe structures [22,23]. In addition, in patient 1, we performed radiofrequency ablation from the mesial temporal lobe with lateral extension to the neocortex that caused temporal stem partial destruction. This approach, theoretically, has a disconnection effect imitating multiple subpial transection, which stops the generalized propagation of epileptic discharges [24]. To salvage CC in patient 2, we placed greater emphasis on targeting than on volume. In this scenario, robotic guidance was helpful to set the ablation trajectories within the callosal fibers, and thermal simulation with temperature gradient in particular avoided injuries to crucial vessels nearby, such as the pericallosal arteries or the vein of Galen.

Seizures secondary to intracranial space-occupying lesions such as AVM can be controlled after surgical resection [25]. In our patient 3, however, SEEG confirmed that the irritative tissues around the origin nidus were actually the epileptic foci, and lesioning on them resulted in good seizure control. Preoperative simulation was helpful to avoid injury on the residual vein. However, we did not perform concurrent electrophysiologic guidance because the depth electrodes required for one-stage monitoring and RFTC were not available in our region. Nevertheless, compared to SEEG-guided RFTC with bipolar ablation, which induces an oval shape lesion, our unipolar RARFA would make ablation volume more calculable with a simple mathematic equation.

Regarding traditional craniotomies, cortex traction or manipulation can induce postoperative early seizures [26]. Moreover, repeat craniotomies can potentially cause perioperative complications, especially for deep structures such as the hippocampus or corpus callosum. There were no immediate postoperative seizures nor clinical complications in our series, proving that radiofrequency ablation is a safe treatment which can avoid brain irritation from secondary craniotomies. The two adult patients had improved neuropsychologic function due to good seizure control, as well as no collateral damage to the brain functional area. In addition, our patients had nearly no pain postoperatively and a minimal length of hospital stay. Another available minimally invasive modality, radiosurgery, ideally has the same advantage mentioned above. However, our RARFA protocol had superiority in providing “immediate” seizure remission, while radiosurgery treatment results in a “gradual” increase in seizure control [27].

## 5. Conclusions

In this pilot study, we preliminarily verified the feasibility and safety of a novel S-RARFA protocol based on traditional RFTC. We demonstrated that S-RARFA can serve as a successful seizure reoperation with high accuracy, minimal collateral damage, and good seizure control. Through sharing our experience, we hope that this procedure can increase patient acceptance, as well as epileptologists’ reference to surgeries, since epilepsy surgery has been underutilized for decades.

## Figures and Tables

**Figure 1 jcm-11-04804-f001:**
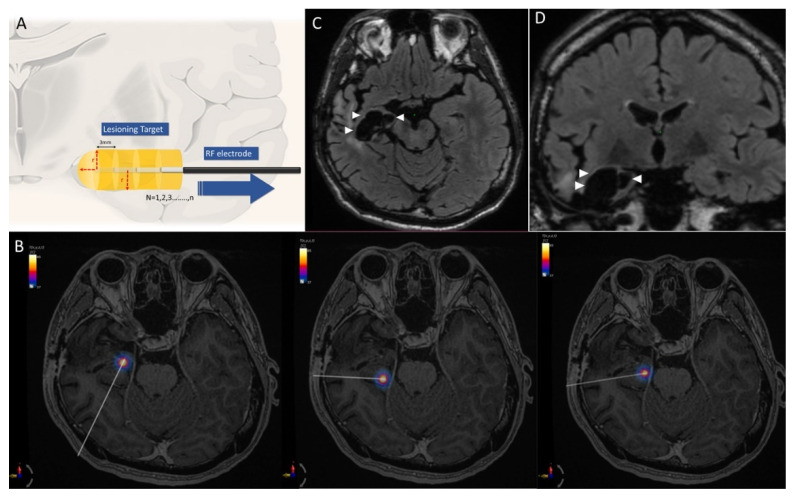
(**A**) Schematic demonstration of RARFA. The ablation trajectory was planned and then started from the deepest target point. The electrode was then pulled backward axially using the robot-assisted micromove program (blue arrow). Each RARFA created a hemisphere at the tip plus a 3-mm-long cylindrical lesion, and the radius depended on a thermal boundary of 60 °C (red dashed line). All lesions comprised the treatment area (yellow). (**B**) Three RARFA trajectories projected on axial MRI (not in the same plane) with thermodynamic simulation of the three deepest ablation points for patient 1. The core heating temperature was around 80 °C, the atmosphere body temperature was 37 °C, and the diameter of the 60 °C boundary was 11 mm. (**C**) Axial and (**D**) coronary MRI images at 18 months post-operation revealed that remnant right hippocampus and parahippocampal gyrus were removed (arrow heads).

**Figure 2 jcm-11-04804-f002:**
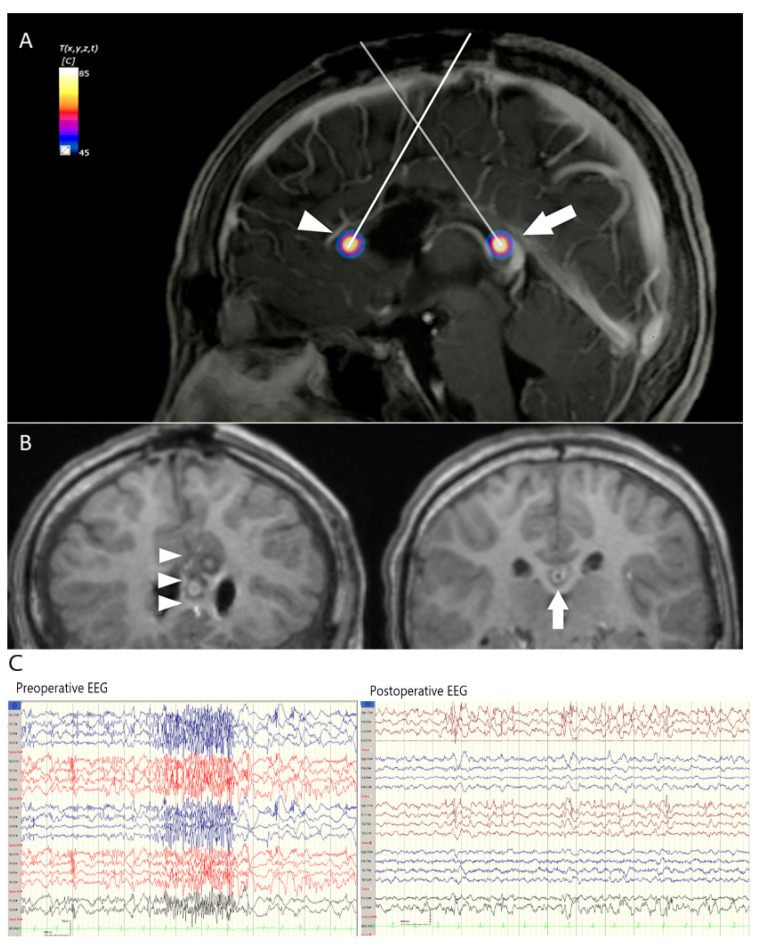
(**A**) Two trajectories to the corpus callosum genu (arrowhead) and splenium (arrow) projected on sagittal MRI with thermodynamic simulation of the deepest point for patient 2. Note that the anterior cerebral arteries (arrowhead) and the vein of Galen (arrow) were outside the 60 °C margin. (**B**) Postoperative MRI showed lesions in the remnant corpus callosum genu and splenium. The right anterior cerebral artery (arrow) was spared, and the posterior lesion safely stayed in the splenium (arrowhead). (**C**) Postoperative EEG showed a significant decrease of generalized paroxysmal fast activity after the surgery, compared to preoperative EEG.

**Figure 3 jcm-11-04804-f003:**
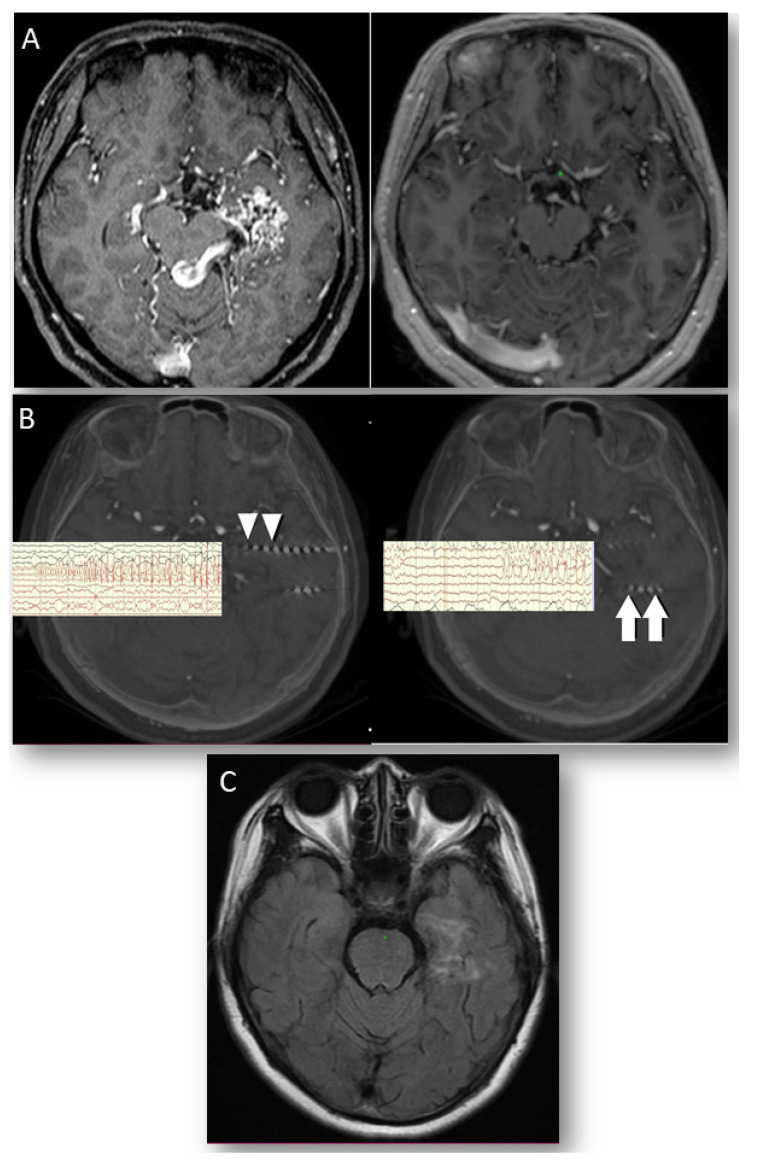
**Figure 3**. (**A**) Left mesial temporal AVM nearly totally obliterated 2 years after radiosurgery with a small residual vein. (**B**) Depth electrodes capture epileptic form discharges over the area anterior (arrowheads) and posterior to the original AVM (arrows). (**C**) T1 weighted MRI at 6 months postoperation revealed a lesion area by RARFA without injury to the residual vein.

**Table 1 jcm-11-04804-t001:** Clinical information. RARFA: robot-assisted radiofrequency ablation; ILAE: International League Against Epilepsy; OP: operation.

Patient	Gender	Age	Underlying Etiology	Previous Surgery	RARFA Target	RARFA Trajectories	Ablation Volume (mm^3^)		Post-OP Pain Score	Post-OP Length of Stay	Post-OP Acute Seizure	ILAE Classification
							Simulation and Calculation	Post-OP Image Measurement (Percentage Error of with Calculation)				
1	Male	28	Right hippocampal sclerosis	Partial anterior temporal lobectomy	Right remanent hippocampus	3	9949.69	9727 (2.3%)	1	2	none	1a
2	Male	9	Lennox Gastaut syndrome	Anterior callosotomy	Corpus callosum	2	1867.87	1841 (1.4%)	1	4	none	4
3	Female	45	Left mesial temporal AVM	Stereotactic radiosurgery	Left mesial temporal (around residual vein)	2	3941.89	4073 (3.2%)	1	3	none	1a

## Data Availability

Not applicable.

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
