# Peer review of "Robot-Assisted Radiofrequency Ablation Combined with Thermodynamic Simulation for Epilepsy Reoperations"

_jcm, 2022, doi:10.3390/jcm11164804_

Round 1

Reviewer 1 Report

Reviewer Comments to Paper “Robot-assisted Radiofrequency Ablation Combined with Thermodynamic Simulation for Epilepsy Reoperations”

Apart from a few grammar inaccuracies (mostly verbal agreement), the paper is quite interesting, with cutting-edge contents that should be made available to professionals dealing with epilepsy surgery

Nervous tissue ablation is an accepted method for treating pharmacoresistant epilepsy, particularly in patients who already failed previous resective or disconnective surgery. Ablative methods include laser interstitial thermal therapy (LiTT) and radiofrequency thermocoagulation (RT), among others. While the first one allows for real-time assessment of the volume of ablated tissue, the latter, albeit controlled, does not have this feature. The present paper aims to overcome such difficulty by using modern tools (robotic surgery along with a thorough preoperative planning using dedicated softwares for computer simulation) and describes the authors’ experience with these tools within a protocol for radiofrequency thermocoagulation. Their method is certainly ingenious and might help other centers perform these procedures more safely, obviously those whose do not have LiTT available. Data presented herein are quite interesting and the clinical outcome of patients is satisfactory, which makes this a promising surgical alternative. On the other hand, only four patients were included, and therefore a larger case series is required to clearly confirm its efficacy. Likewise, many of the equipments required (surgical robots, for instance) are not widespread, so clinical application might be limited. Nevertheless, it is a novel technique that should be made known to a wider public.

Also, I have attached the manuscript file with some grammatical corrections within the running text, to make it more accessible.

Author Response

Apart from a few grammar inaccuracies (mostly verbal agreement), the paper is quite interesting, with cutting-edge contents that should be made available to professionals dealing with epilepsy surgery

Nervous tissue ablation is an accepted method for treating pharmacoresistant epilepsy, particularly in patients who already failed previous resective or disconnective surgery. Ablative methods include laser interstitial thermal therapy (LiTT) and radiofrequency thermocoagulation (RT), among others. While the first one allows for real-time assessment of the volume of ablated tissue, the latter, albeit controlled, does not have this feature. The present paper aims to overcome such difficulty by using modern tools (robotic surgery along with a thorough preoperative planning using dedicated softwares for computer simulation) and describes the authors’ experience with these tools within a protocol for radiofrequency thermocoagulation. Their method is certainly ingenious and might help other centers perform these procedures more safely, obviously those whose do not have LiTT available. Data presented herein are quite interesting and the clinical outcome of patients is satisfactory, which makes this a promising surgical alternative. On the other hand, only four patients were included, and therefore a larger case series is required to clearly confirm its efficacy. Likewise, many of the equipments required (surgical robots, for instance) are not widespread, so clinical application might be limited. Nevertheless, it is a novel technique that should be made known to a wider public.

Also, I have attached the manuscript file with some grammatical corrections within the running text, to make it more accessible.

We deeply appreciated your compliment to our work and your grammar correction to make this paper better. In addition, we concluded all reviewers’ comment and decided to delete demonstration of case 4. Instead, we descripted more on seizure profile and presurgical work up for the other three cases. We believed that would make our readers no confusion about the advantage of RARFA. This manuscript is “communication” type, three cases would be enough to share the preliminary experience of S-RARFA, and we would recruit more cases in the future.

We didn’t revise the words in conclusion as your suggestion because we thought the origin sentence would be more proper.

Thank you again for your review and comment.

Reviewer 2 Report

Dear Authors,

the aim of the paper is clearly and appropriately defined (the reader can only desume it). The paper is moving between the usefulness of softwares to predict lesion size and the clinical outcome of this application. It's not clear which is the aim, again. This is the first critic I do and I will not go ahead with furthers (I'd have many), since a paper needs clarity of aim and, consequently, conclusion, in my opinion. 

best regards

Author Response

the aim of the paper is clearly and appropriately defined (the reader can only desume it). The paper is moving between the usefulness of softwares to predict lesion size and the clinical outcome of this application. It's not clear which is the aim, again. This is the first critic I do and I will not go ahead with furthers (I'd have many), since a paper needs clarity of aim and, consequently, conclusion, in my opinion.

First, we apologized to make you misunderstanding. We took this manuscript as sort of “surgical technical note” rather than a standard clinical study; which means, we presented a protocol for mini-invasive seizure surgery and shared our pilot experience to the readers. That’s the major expectation for the “brief report” or “communication” type of paper of this journal.

Please let us explain more.

Currently, there are two kinds of ablative methods to treat seizure including laser interstitial thermal therapy (LiTT) and radiofrequency thermocoagulation (RFTC). While the first one allows for real-time assessment of the volume of ablated tissue, the latter does not have this feature. We didn’t have the modern equipment as LiTT in our institute but only RFTC. The present study aims to overcome such difficulty, and we adapted a protocol with the modalities we have (simulation program and robotic assistance). Actually, the procedure contained” doing more with less” concept.

Like you mentioned above, the usefulness of computer simulation as well as robotic stereotactic assistance to make precise lesioning size and good surgical outcome are both our aims, which are not conflict to each other.

We believed that in this short report, we clearly described our clinical experience accentuate the advantage containing computational and robotic tools within a protocol for seizure surgery. We deleted case 4 according to other reviewers comment to make readers no confusion.

Last but not least, we revised the introduction and conclusion section to clearly describe the necessary information and specify our purpose.

We deeply appreciate your comment. Hope this would answer your concern and meet your request.

Thank you again.

Reviewer 3 Report

(General)

Surgical approaches for refractory epilepsy refractory to many oral epilepsy medications, but with recurrent seizures, have been very limited.
Although the number of cases reported by the authors is small (one pediatric and three adult cases), their report is an amazing case of epilepsy surgery that includes new findings.

   While minimally invasive radiofrequency ablation is an alternative therapy, the authors used simulation and robot-assisted radiofrequency ablation (S-RARFA) protocols to plan and perform cerebral epileptic tissue lesions.
In other words, the detection of epileptic foci in the peri-lesional stimulated areas such as residual hippocampus, corpus callosum avulsion, and arteriovenous malformations, as well as arachnoid cysts, all can be surgically treated with S-RARFA. The relationship between preoperative simulations and postoperative measurements has also been evaluated, and the postoperative course has been excellent.
The authors conclude that this novel S-RARFA therapy served as an alternative to conventional surgery and real-time monitoring to successfully re-operate on seizures with minimal collateral injury.

(Introduction) 

In the first half of the introduction, I would like the journal, being a general medical journal, to give a broad description of epilepsy in general and then delve into the history of surgical treatment.
   Therefore. After describing the definition, frequency, and general treatment of epilepsy, please describe which cases are suitable for epilepsy surgical treatment, which cases are considered refractory epilepsy as well as temporal lobe epilepsy, and what epilepsy surgical techniques have generally been recommended on that basis.
   The authors should also include how many or more antiepileptic drugs they consider cases of refractory epilepsy to be refractory to seizure control.
A few descriptions of epilepsy associated with symptomatically severe physical and mental disorders should also be included.
   We would then ask that the authors revise their description of surgical treatment to include details.

(Materials and Methods)

Please consider whether the following should be included in this section.
In each individual case, what tests were performed before deciding to proceed with epilepsy surgery?
   It would be desirable to have an additional statement as to whether your hospital's epilepsy multidisciplinary team (what specific specialists are involved?) determined that surgical treatment was necessary on that basis.
   Is an intelligence test (IQ) performed prior to epilepsy surgery?
   Is the Wada test, for example, performed to evaluate the dominant hand?
   In addition to MRI, is nuclear medicine SPECT (IMP, ECD, IMZ) performed as a cerebral blood flow test? IMP, ECD, IMZ, etc.) and FDG and FDG, flumazenil, etc. PET scan? It would be desirable to have a more neuro-radiological medical description.

The epilepsy surgical description was detailed and well understood.
Would it be possible to add a picture of the surgical equipment system as a diagram as an addition to the setting?

(Results)

The four cases shown in the results, three adults and one child, are all varied, varied, and informative.

The tables are easy to read and understood.

The presentation of the four cases presented is very interesting.

Case1 is OK, but in Case 2, how long was the postoperative course of Lennox-Gastaut syndrome in case 2 observed? Even if the drop attacks disappeared after the surgery of How long was the postoperative course of Lennox-Gastaut syndrome in case 2 observed? Even if the drop attacks disappeared after the cerebral surgery to the corpus callosum , there is a high probability that partial seizures on either side would remain. We would appreciate confirmation on this point. We would also like to know how the course of oral anti-epileptic drug therapy changed after the surgery., there is a high probability that partial seizures on either side would remain. We would appreciate confirmation on this point. We would also like to know how the course of oral anti-epileptic drug therapy changed after the surgery.

Case 3 was well understood.
Was the arachnoid cyst in case 4 really related to epileptic seizures? Were there any imaging findings of brain tissue compression and drainage? Also, were there any accompanying symptoms such as vomiting with the seizures?

Good description according to ILAE standards.

(Discussion and Conclusion)

Overall, a high level of consideration is presented. Please review the revised description so far and revise where necessary.

The reviewers have expressed interest in this paper, look forward to your re-submission.

Best regards,

Reviewer

Author Response

Surgical approaches for refractory epilepsy refractory to many oral epilepsy medications, but with recurrent seizures, have been very limited.
Although the number of cases reported by the authors is small (one pediatric and three adult cases), their report is an amazing case of epilepsy surgery that includes new findings.

   While minimally invasive radiofrequency ablation is an alternative therapy, the authors used simulation and robot-assisted radiofrequency ablation (S-RARFA) protocols to plan and perform cerebral epileptic tissue lesions.
In other words, the detection of epileptic foci in the peri-lesional stimulated areas such as residual hippocampus, corpus callosum avulsion, and arteriovenous malformations, as well as arachnoid cysts, all can be surgically treated with S-RARFA. The relationship between preoperative simulations and postoperative measurements has also been evaluated, and the postoperative course has been excellent.
The authors conclude that this novel S-RARFA therapy served as an alternative to conventional surgery and real-time monitoring to successfully re-operate on seizures with minimal collateral injury.

(Introduction) 

In the first half of the introduction, I would like the journal, being a general medical journal, to give a broad description of epilepsy in general and then delve into the history of surgical treatment.
   Therefore. After describing the definition, frequency, and general treatment of epilepsy, please describe which cases are suitable for epilepsy surgical treatment, which cases are considered refractory epilepsy as well as temporal lobe epilepsy, and what epilepsy surgical techniques have generally been recommended on that basis.
   The authors should also include how many or more antiepileptic drugs they consider cases of refractory epilepsy to be refractory to seizure control.
A few descriptions of epilepsy associated with symptomatically severe physical and mental disorders should also be included.
   We would then ask that the authors revise their description of surgical treatment to include details.

  • Thank you for your advice. We expanded the introduction section to describe 1. the ILAE definition of epilepsy; 2. drug refractory epilepsy (failure to 2 more AEDs), in which surgical intervention should be considered; 3. surgical resection or ablation is indicated for focal seizure, while disconnecting surgery like corpus callosotomy is suitable for patients with wide spreading, generalized seizure; 4. the Lennox Gastaut syndrome since this is relevant to our case demonstration.

(Materials and Methods)

Please consider whether the following should be included in this section.
In each individual case, what tests were performed before deciding to proceed with epilepsy surgery?
   It would be desirable to have an additional statement as to whether your hospital's epilepsy multidisciplinary team (what specific specialists are involved?) determined that surgical treatment was necessary on that basis.
   Is an intelligence test (IQ) performed prior to epilepsy surgery?
   Is the Wada test, for example, performed to evaluate the dominant hand?
   In addition to MRI, is nuclear medicine SPECT (IMP, ECD, IMZ) performed as a cerebral blood flow test? IMP, ECD, IMZ, etc.) and FDG and FDG, flumazenil, etc. PET scan? It would be desirable to have a more neuro-radiological medical description.

The epilepsy surgical description was detailed and well understood.
Would it be possible to add a picture of the surgical equipment system as a diagram as an addition to the setting?

  • We added the description of preoperative workup including scalp EEG and PET-MRI, which were for electrophysiologic, anatomic and metabolism assessment. We also mentioned that the member of multidisciplinary team included epileptologist, neuroradiologist, neuropsychologist, and neurosurgeon.
  • IQ test were performed in adult patients using the Wechsler Adult Intelligence Scale to evaluate memory function and IQ, and we put the result into the table. Wada test is no longer performed as seizure surgery routine in our institution.

We put the diagram of surgical concept as figure 1a, and the videos were clear to showed the ablation procedure. We didn’t put the surgical equipment as ROSA in the figure to avoid conflict of interest; and because readers would find one on website easily.

(Results)

The four cases shown in the results, three adults and one child, are all varied, varied, and informative.

The tables are easy to read and understood.

The presentation of the four cases presented is very interesting.

Case1 is OK, but in Case 2, how long was the postoperative course of Lennox-Gastaut syndrome in case 2 observed? Even if the drop attacks disappeared after the surgery of How long was the postoperative course of Lennox-Gastaut syndrome in case 2 observed? Even if the drop attacks disappeared after the cerebral surgery to the corpus callosum , there is a high probability that partial seizures on either side would remain. We would appreciate confirmation on this point. We would also like to know how the course of oral anti-epileptic drug therapy changed after the surgery., there is a high probability that partial seizures on either side would remain. We would appreciate confirmation on this point. We would also like to know how the course of oral anti-epileptic drug therapy changed after the surgery.

Case 3 was well understood.
Was the arachnoid cyst in case 4 really related to epileptic seizures? Were there any imaging findings of brain tissue compression and drainage? Also, were there any accompanying symptoms such as vomiting with the seizures?

Good description according to ILAE standards.

  • Case 2: we treated this patient 7 years before the operation and one and half year postoperatively (so far to the time of manuscript submission). Like we mentioned in the introduction, the purpose of disconnecting surgery like corpus callosotomy is not to cease the generalized seizure (it’s impossible) but to stop the epileptiform discharge spreading globally, which causes drop attack and impairs life mostly. He had previous incomplete callosotomy and the symptoms recurred soon after the operation. Therefore, through RARFA, we stopped this most bothering symptom in this LGS patient and improved quality of life both for him and his family postoperatively. Moreover, the generalized paroxysmal fast activity significantly decreased after the surgery. (showed in revised figure 2c). Therefore, the kid took 4 anti-epileptic drugs currently, compared to 5 AEDs prior to surgery. We added description in this case demonstration and figure caption.
  • We deleted case 4 demonstration according to your and other reviewers’ concern. We would like to make no confusion to readers. Since this is a “communication” type of manuscript, we thought 3 cases would be enough to show the pilot experience of our protocol.

(Discussion and Conclusion)

Overall, a high level of consideration is presented. Please review the revised description so far and revise where necessary.

The reviewers have expressed interest in this paper, look forward to your re-submission.

We appreciated your suggestion again. Hope the revision would answer your concern.

Round 2

Reviewer 2 Report

Dear Authors,

                        I confirm my previous doubt and I do not consider sufficient your rebuttal. As stated before, I could forward you other critics (clinical, not only methodological), such as the use of RF-THC for a residual AVM...

The methodology is interesting but first I would perform a methodological paper, and then a clinical one on grounded basis.

best regards

Author Response

Dear Authors,

I confirm my previous doubt and I do not consider sufficient your rebuttal. As stated before, I could forward you other critics (clinical, not only methodological), such as the use of RF-THC for a residual AVM...

The methodology is interesting but first I would perform a methodological paper, and then a clinical one on grounded basis.

best regards

Dear reviewer

  • The utilization of RFTC as a minimally invasive seizure surgery has been well established and published in previous studies, which were cited in the reference7,8,9,12. Besides, robot stereotactic assistance (ROSA) is a medically commercialized device and has been approval for clinical use globally (US-FDA, Europe-CE, and TFDA). Therefore, strictly speaking, we only added preoperative computer simulation to overcome the lack of real-time temperature monitoring during ablation. There were animal models on this simulation technique, and the results showing its accuracy were cited in reference 13, 14. In this communication paper, we clarify that our purpose is to demonstrate the surgical protocol as an alternative to traditional method. We hope that would be helpful for other institutes where no LiTT available, and we encouraged them to reproduce and verify the protocol. We echoed what you said, this is actually a methodological paper. Would you please give us a chance to share this pilot experience?
  • The protocol has been approved by IRB, and all the clinical cases we recruited has been reviewed by a multi-disciplinary team including epileptologists, neuroradiologists, neuropsychologists and neurosurgeons; which means, the potential feasibility and safety of the protocol have been assessed groundedly.
  • In case 3, we didn’t use RARFA to ablate “residual AVM” but the surrounding epileptic tissue, which was confirmed by SEEG first. More importantly, it is impossible for us to do thermocoagulation directly on the vascular lesion. The truth is, we tried to avoid it. That’s exactly the usefulness of thermodynamic simulation, minimizing the risk to cause residual vein bleeding by thermocoagulation injury.

Reviewer 3 Report

I appreciate this major revision of your paper.

The introduction has been changed to make it easier to understand for readers with limited knowledge of epilepsy.
It is also easier to understand for readers who have little interest in or are new to epilepsy surgery.

I was interested in epilepsy surgical treatment of arachnoid cysts. However the removal of Case 4 has been boldly done to make the paper more consistent.

The MRI and EEG images are also appreciated and high qualities.

The two additional video clips are also very interesting viewing.

In general, the contributors' revisions of the this interesting epileptic surgery for 3 cases of intractable epileptic report were properly addressed in response to all reviewers' questions.
So, this paper should be acceptable for the Journal of Clinical Medicine.

Best regards,

Reviewer

Author Response

I appreciate this major revision of your paper.

The introduction has been changed to make it easier to understand for readers with limited knowledge of epilepsy.
It is also easier to understand for readers who have little interest in or are new to epilepsy surgery.

I was interested in epilepsy surgical treatment of arachnoid cysts. However the removal of Case 4 has been boldly done to make the paper more consistent.

The MRI and EEG images are also appreciated and high qualities.

The two additional video clips are also very interesting viewing.

In general, the contributors' revisions of the this interesting epileptic surgery for 3 cases of intractable epileptic report were properly addressed in response to all reviewers' questions.
So, this paper should be acceptable for the Journal of Clinical Medicine.

  • We really appreciated your comment. Removal of the case 4 demonstration is a difficult decision for us, but we want to make the paper more soild, like you said. We would recruit more patients in the future to the S-RARFA treatment and descript the arachnoid cyst case with a more proper demonstration.

Thank you again.
